



# Modeling Commercial-Scale CO₂ Storage in the Gas Hydrate Stability Zone with PFLOTRAN v6.0

Michael Nole[1], Jonah Bartrand[1], Fawz Naim[2], and Glenn Hammond[1]

[1]Pacific Northwest National Laboratory, Richland, 99354, USA
[2]Ohio State University, Columbus, 43210, USA

*Correspondence to*: Michael Nole (michael.nole@pnnl.gov)

**Abstract**

Safe and secure carbon dioxide ($CO_2$) storage is likely to be critical for mitigating some of the most dangerous effects of climate change. In the last decade, there has been a significant increase in activity associated with reservoir characterization and site selection for large-scale $CO_2$ storage projects across the globe. These prospective storage sites tend to be terrestrial sites selected for their optimal structural, petrophysical, and geochemical trapping potential. However, it has also been suggested that storing $CO_2$ in reservoirs within the gas hydrate stability zone (GHSZ), characterized by high pressures and low temperatures (e.g., Arctic or marine environments), could provide natural thermodynamic and solubility barriers to gas leakage. Evaluating the prospect of commercial-scale, long-term storage of $CO_2$ in the GHSZ requires reservoir-scale modelling capabilities designed to account for the unique physics and thermodynamics associated with these systems. We have developed the HYDRATE flow mode and accompanying fully implicit parallel well model in the massively parallel subsurface flow and reactive transport simulator PFLOTRAN to model $CO_2$ injection into the marine GHSZ. We have applied these capabilities to a set of $CO_2$ injection scenarios designed to reveal the challenges and opportunities for commercial-scale $CO_2$ storage in the GHSZ.

## 1 Introduction

Large-scale deployment of carbon capture and storage (CCS) projects is likely to be critical for constraining future global temperature increase due to climate change, yet major uncertainties exist regarding potential injectivity of $CO_2$ in subsurface reservoirs (Lane et al., 2021). Evaluating terrestrial $CO_2$ storage sites for long-term sequestration requires synthesizing sophisticated laboratory, field, and modelling tools to assess the $CO_2$ trapping potential of a prospective subsurface reservoir during a large-scale injection over a significant post-injection performance period. $CO_2$ trapping in terrestrial sequestration involves potentially both physical and chemical trapping mechanisms which include structural, solubility, capillary, and mineralization trapping (Al Hameli et al., 2022). Currently, large-scale carbon capture and storage (CCS) projects around the globe are each storing over 400,000 metric tons (Mt) of $CO_2$ annually (Snæbjörnsdóttir et al., 2020); the U.S. Department of Energy's CarbonSAFE initiative aims to develop $CO_2$ storage complexes across the U.S. that would be capable of storing total





volumes exceeding 50 million metric tons (MMt) of $CO_2$ each (Sullivan et al., 2020). To achieve this vision, a diverse set of potential reservoir host rocks and environments is being considered. This includes speculation about the feasibility of offshore $CO_2$ sequestration and mineralization, such as in Cascadia Basin basalts offshore the U.S. Pacific Northwest (Goldberg et al., 2018). Shallow sub-sea environments are not only isolated from the atmosphere by a large water body, but they can also exist in a unique pressure and temperature regime conducive for forming gas hydrate.


    Gas hydrate is a solid-phase, non-stoichiometric mixture of low molecular weight gas molecules occupying free spaces in a solid water lattice. Hydrates of several different gases occur abundantly in nature, but since the hydrate phase is only stable at high pressures and low temperatures it is only found naturally on Earth in the pore space of soils in either permafrost or sub-sea environments. Methane hydrate is of interest for its potential as a natural gas energy resource (Collett, 2000; Oyama et al.,

2017; Singh et al., 2022), for its potential role in global carbon cycling as the climate changes (Ruppel and Kessler, 2017), and for its role as a geohazard (Zander et al., 2018; Kaminski et al., 2020). On the other hand, carbon dioxide ($CO_2$) hydrate, which forms at similar pressures and temperatures as methane hydrate, is increasingly being explored as a potential means to permanently sequester $CO_2$ as a climate change mitigation solution that comes with additional safety factors beyond those typically encountered in terrestrial $CO_2$ sequestration scenarios. These include the fact that it is immobile in sediment pore

space, which adds a thermodynamic barrier to gas escape, and the presence of a significant ocean sink in marine environments which isolates $CO_2$ from release into the atmosphere (Tohidi et al., 2010). An added benefit of $CO_2$ injection into the methane hydrate stability zone is that $CO_2$ hydrate is typically more thermodynamically favourable than methane hydrate, meaning it could theoretically be possible to use $CO_2$ to kick out methane from the hydrate phase, thus sequestering $CO_2$ while producing natural gas from a methane hydrate deposit (Koh et al., 2016).


    $CO_2$ sequestration in gas hydrate form can only occur in a finite bounded temperature and pressure range. In terms of a soil column, there exists a depth-bounded gas hydrate stability zone (GHSZ) in the subsurface in which hydrate can form. In a marine environment, the GHSZ typically begins several meters above the seafloor, but hydrate does not form freely in the water column (except for, e.g., as a gas bubble crust [Fu et al., 2021]) because the guest molecule gas (e.g., $CO_2$) typically

cannot become concentrated enough in the water to do so. Therefore, the seafloor is typically the shallowest extent of hydrate formation in marine systems. Relative to deeper sediments, this point usually represents a minimum temperature and pressure. Working downward through the GHSZ, pressure increases roughly hydrostatically and temperature increases along a geothermal gradient. Increases in pressure stabilize hydrate, while increases in temperature destabilize hydrate. The geothermal temperature change effect on hydrate stability outweighs the hydrostatic pressure change effect, so there exists a depth below

the seafloor where the temperature is too high to form hydrate, known as the base of the gas hydrate stability zone (BHSZ). Overall, the specific thickness of the bulk GHSZ is dependent on pressure, temperature, porewater salinity and gas composition (Sloan and Koh, 2007).





For a potential host reservoir within the GHSZ, the long-term $CO_2$ storage potential of the reservoir would consider the
thermodynamic trapping mechanism of solid gas hydrate formation in addition to traditional trapping mechanisms. Several
experimental studies have demonstrated the process of $CO_2$ trapping and hydrate conversion in the GHSZ at the lab-scale,
demonstrating how conversion of $CO_2$ into a solid phase adds an additional safety factor (Gauteplass et al., 2020; Rehman et
al., 2021). An experimental study of layered sediments using different injection strategies demonstrated the need to consider
thermal management when designing a $CO_2$ injection in the GHSZ and suggested multilateral perforated horizontal wells may
achieve the most optimal $CO_2$ conversion efficiency (Pang et al., 2024). However, reservoir-scale modelling studies of the
transport and thermodynamic phenomena associated with injection of $CO_2$ in commercial volumes into the GHSZ are lacking.

We present several new capabilities developed in the open source, massively parallel multiphase flow and reactive transport
simulator PFLOTRAN (Hammond et al., 2014) to model reservoir-scale injection of $CO_2$ in the GHSZ. We have extended
PFLOTRAN's HYDRATE mode capabilities to model free-phase $CO_2$ flow properties and $CO_2$ hydrate phase behaviour.
Additionally, we introduce a fully coupled parallel well model that can be used to model $CO_2$ injection into heterogeneous
media and can adapt to changes in flow properties associated with hydrate formation in the vicinity of the wellbore. Finally,
we add a new fully coupled salt mass balance to consider salinity and salt precipitation effects in the GHSZ. We demonstrate
these capabilities on a series of test problems designed to elucidate the challenges and opportunities associated with
commercial-scale injection of $CO_2$ into the GHSZ.

## 2 Methods

PFLOTRAN's HYDRATE mode was originally developed to model methane generation, transport, and structure 1 (SI) gas
hydrate formation in deep marine and Arctic terrestrial reservoirs. PFLOTRAN's HYDRATE mode has been benchmarked
against other reservoir simulators for modelling methane gas production from hydrate reservoirs (White et al., 2020). It has
been used to predict shallow gas generation and gas hydrate formation offshore the eastern U.S. (Eymold et al., 2021), to study
relationships between gas generation and slope stability along the U.S. Atlantic margin (Carty and Daigle, 2022), and to model
gas hydrate accumulation offshore Norway (Frederick et al., 2021). An extension of HYDRATE mode to include salinity
coupling was developed to investigate viscous fingering and convective mixing in layered marine sediments during methane
hydrate formation over geologic time (Fukuyama et al., 2023). Here, we have redeveloped PFLOTRAN's HYDRATE mode
to optionally consider $CO_2$ as the working gas; to couple fully implicitly with a new parallel well model; to include a new fully
coupled salt mass balance; and to consider variable salinity effects on $H_2O$-$CO_2$-NaCl mixtures and the $CO_2$ hydrate phase
boundary.





## 2.1 Governing Equations

A system of three mass balance equations, one energy balance, and one well equation is now solved fully implicitly in PFLOTRAN's HYDRATE mode. The mass conservation equations take the following form:

$$\frac{\partial}{\partial t}\phi \sum_{\alpha=\mathrm{l,g,h,i,s}}\left(s_\alpha \rho_\alpha x_j^\alpha\right) + \nabla\cdot\left(\boldsymbol{q}_\mathrm{l}\rho_\mathrm{l}x_j^\mathrm{l} + \boldsymbol{q}_\mathrm{g}\rho_\mathrm{g}x_j^\mathrm{g} - \phi s_\mathrm{l}D_\mathrm{l}\rho_\mathrm{l}\nabla x_j^\mathrm{l} - \phi s_\mathrm{g}D_\mathrm{g}\rho_\mathrm{g}\nabla x_j^\mathrm{g}\right) = Q_j + Q_{\mathrm{w},j}\,, \tag{1}$$

where phase $\alpha$ can be liquid (l), gas (g), hydrate (h), ice (i), or salt precipitate (s); component $j$ includes water, gas ($CO_2$, $CH_4$, or air), and salt (NaCl); $s_\alpha$ is the saturation of phase $\alpha$; $\rho_\alpha$ is the density of phase $\alpha$; $x_j^\alpha$ is the mole fraction of component $j$ in phase $\alpha$; $q_\mathrm{l}$ is the liquid Darcy flux vector; $q_\mathrm{g}$ is the gas Darcy flux vector; $D_\mathrm{l}$ is the liquid phase diffusivity; $D_\mathrm{g}$ is the gas phase diffusivity; $\phi$ is the porosity; and $Q_j$ includes any non-well sources/sinks of component $j$; and $Q_{\mathrm{w},j}$ is a source/sink of component $j$ from a well. Solid phases are considered immobile and include the hydrate, ice, and salt precipitate phases. Mole fractions of components in the solid phases are fixed: by the hydration number in the hydrate phase, as pure water in the ice phase, and as pure salt in the salt precipitate phase. Formation of gas hydrate and ice therefore results in salt exclusion and aqueous dissolved salinity enhancement, which affects the hydrate phase boundary and gas solubility in the brine.

The energy conservation equation takes the form:

$$\sum_{\alpha=\mathrm{l,g,h,i,s}}\left(\frac{\partial}{\partial t}(\phi s_\alpha \rho_\alpha U_\alpha) + \nabla\cdot(\boldsymbol{q}_\alpha \rho_\alpha H_\alpha)\right) + \frac{\partial}{\partial t}\left((1-\phi)\rho_\mathrm{r}C_p T\right) - \nabla\cdot(\kappa\nabla\mathrm{T}) = Q_e + Q_{\mathrm{w},e}\,, \tag{2}$$

where $U_\alpha$ is the internal energy of phase $\alpha$, $H_\alpha$ is the enthalpy of phase $\alpha$, $\rho_\mathrm{r}$ is the rock density, $C_p$ is the heat capacity of the rock, $\kappa$ is the composite thermal conductivity of the medium, $T$ is the temperature, $Q_e$ includes any non-well heat sources/sinks, and $Q_{\mathrm{w},e}$ is a heat source/sink imposed by the well (e.g., a heater in addition to a fluid injection). Exothermic hydrate formation (and vice versa, i.e., endothermic hydrate dissociation) is captured here by a decrease in internal energy of the hydrate phase during formation; this typically results in either an increase in system temperature or a change in phase saturations in three-phase systems. As we will show later, this phenomenon is important during $CO_2$ injection in the short term and it can continue to buffer conversion between phases for hundreds of years; similar effects have been shown for natural $CH_4$ hydrate systems where the base of the gas hydrate stability zone is shifted due to climactic changes (Owulunmi et al., 2022).

A fully implicit, parallel well model has also been incorporated into HYDRATE mode. A well model can more accurately represent the insertion of a (comparably) small cylindrical wellbore into a reservoir grid cell than a standard source/sink term. Given a prescribed surface injection rate of $CO_2$ into the well, the well model solves for pressure variation along a wellbore





and dynamically adjusts flow rates into the reservoir in response to changes in reservoir physical properties like permeability.
This phenomenon can be critical to capture in a horizontal well or injection into a heterogeneous reservoir in the gas hydrate stability zone, where near-wellbore formation (or dissociation) of gas hydrate can significantly alter reservoir permeability and thus injection behaviour. The well model developed for HYDRATE mode is a hydrostatic well model based off the design of White et al. (2013) but with key modifications including full parallelization to run flexibly on very large, unstructured grids and the addition of a thermal component; as we show here, injection temperature could be one of the most important design
considerations for $CO_2$ storage in the gas hydrate stability zone. The well model developed here accounts for the enthalpy of the injected $CO_2$ at the prescribed temperature and wellbore pressure using the same equation of state (EOS) as the reservoir.

Solving a hydrostatic well model involves solving one extra conservation equation per well in addition to the reservoir mass and energy conservation equations. This means that for one well, only one extra row and one extra column are used in the fully
implicit flow Jacobian, *not* an extra row and extra column for each reservoir cell associated with a well. For each reservoir cell intersected by a well, well pressure is computed at the centroid of the well section crossing through the reservoir. All well pressures are determined from the bottom hole pressure, a primary variable (see Section 2.3). The well model conservation equation is compact and reads as follows:

$$\sum_i Q_{w,j}^i = q_{w,j}, \tag{3}$$

where $i$ is the discrete reservoir cell index through which the wellbore passes, $Q_{w,j}^i$ are the reservoir source/sink terms of phase $j$ associated with a well in reservoir grid cell $i$, and $q_{w,j}$ is the prescribed surface injection rate of phase $j$.

**2.2 Constitutive Relationships**

Diffusive flux is modelled using Fick's Law with diffusivities computed as functions of temperature and salinity for $CO_2$ (Cadogan et al., 2014) and NaCl (Reid et al., 1987). Advective fluxes of mobile phases are computed by employing a two-phase Darcy's Equation:

$$q_\alpha = -\frac{kk_\alpha^r}{\mu_\alpha}\nabla(P_\alpha - \rho_\alpha \boldsymbol{g}z) \tag{4}$$


where $k$ is the intrinsic medium permeability, $k_\alpha^r$ is the relative permeability of phase $\alpha$, $\mu_\alpha$ is the viscosity of phase $\alpha$, $P_\alpha$ is the pressure of phase $\alpha$, $\boldsymbol{g}$ is the gravity vector, and $z$ is depth. Relative permeability is computed as a function of phase saturations according to one of a suite of standard relative permeability relationships available in PFLOTRAN. Phase densities and viscosities are computed as function of temperature, pressure, and salinity according to several options in PFLOTRAN;





for CO$_2$, the Span-Wagner equation of state is recommended, and for pure water the IF97 equation of state is typically used. Salt is tracked only in the aqueous phase, and affects brine density (Haas, 1976), viscosity (Phillips et a., 1981), enthalpy, and diffusivity (Cadogan et al., 2014; Belgodere et al., 2015).

Gas phase pressure and liquid phase pressure are related as a function of gas phase saturation through a choice of capillary

pressure functions available in PFLOTRAN. When the gas hydrate phase is present, a capillary pressure associated with the hydrate phase is computed using the same capillary pressure function as the gas phase, scaled by the ratio of interfacial tension vis-à-vis Leverett scaling (Leverett, 1941). This capillary pressure is used in the Gibbs-Thomson equation vis-à-vis the Young-Laplace equation to determine the hydrate 3-phase equilibrium temperature depression required to precipitate hydrate in pores as follows:


$$\Delta T_\mathrm{m} = -\frac{T_\mathrm{mb} P_\mathrm{c}}{\Delta H_\mathrm{m} \rho_\mathrm{h}} \tag{5}$$

where $\Delta T_\mathrm{m}$ is the change in the hydrate melting temperature, $P_\mathrm{c}$ is the hydrate phase capillary pressure, $T_\mathrm{mb}$ is the bulk melting temperature, $\Delta H_\mathrm{m}$ is the specific enthalpy of the phase transition, and $\rho_\mathrm{h}$ is the density of solid hydrate. A similar method is

often used to compute ice freezing temperature depression vis-à-vis the Clausius-Clapeyron equation. This effect is typically only significant in fine-grained sediments and/or at very high effective hydrate phase saturations (Anderson et al., 2003).

When both gas hydrate and free gas occupy significant fractions of the pore space, as would be common during CO$_2$ injection, their combined presence in the pore system should be accounted for through an effective saturation that is passed to the

capillary pressure function. At three-phase (aqueous, free gas/CO$_2$ phase, gas hydrate) equilibrium, the chemical potential of CO$_2$ in the gas hydrate phase at a given hydrate capillary pressure must equal that of CO$_2$ in the free gas phase at a different free gas capillary pressure and dissolved CO$_2$. At bulk thermodynamic equilibrium, free gas and gas hydrate are stable together at a single pressure and temperature. In porous media, capillary effects on both the hydrate phase and gas phase lead to a window of possible pressures and temperatures over which three-phase equilibrium can be maintained (Clennell et al., 1999).

To incorporate this effect and maintain thermodynamic reversibility, we adopt the approach of Liu and Flemings (2011) and require free gas and gas hydrate to partition the large pore space equally when both are present (Nole et al., 2018). This partitioning scheme results in the following effective saturations of free gas and gas hydrate:

$$s_{\alpha,\mathrm{eff}} = \begin{array}{l} 2s_\alpha \,, \quad s_\alpha < s_\beta \\ s_\alpha + s_\beta \,, \mathrm{otherwise} \end{array} \tag{6}$$


where $s_{\alpha,\mathrm{eff}}$ is the effective saturation of nonwetting phase $\alpha$ and $\beta$ is the other nonwetting phase in a 3-phase system where liquid water is the wetting phase.



Well flux at each reservoir grid cell is computed as a function of the pressure difference between the well and the reservoir
cell (free gas [$CO_2$] phase pressure for gas injection) scaled by the well index as follows:

$$Q_{w,j} = \frac{WI\rho_j}{\mu_j}\left(P_w - (P_r + \rho_j \boldsymbol{g}\Delta z_{w-r})\right) \tag{7}$$

where $P_w$ is the well node pressure, $P_r$ is the reservoir pressure of phase $j$ in the grid cell associated with a given well node,
and $\Delta z_{w-r}$ is the vertical distance between well node centre and reservoir cell centre. The well index, $WI$, is calculated using a
3D extension of the Peaceman equation incorporating wellbore radius, well skin factor, reservoir directional permeability, and
reservoir grid discretization (White et al., 2013).

The presence of gas hydrate in the pore space of a reservoir decreases the reservoir's permeability below its intrinsic (water-
saturated) permeability. We model permeability reduction as a function of hydrate saturation as follows (Dai and Seol, 2014):

$$k_{eff} = \frac{(1-s_h)^3}{(1+2s_h)^2} \tag{8}$$

where $k_{eff}$ is the effective permeability coefficient and is multiplied by intrinsic permeability to compute the effective absolute
permeability.

Heat transfer occurs through mobile fluid phase flow, phase transitions, thermal conduction, and injection/production. Fluid
phase enthalpies are computed using corresponding equations of state. For the $CO_2$ phase, the Span-Wagner equation of state
is recommended (Span and Wagner, 1996), and for water the IF97 equation of state with salinity extensions are available in
PFLOTRAN. Enthalpies of the solid gas hydrate (Handa, 1998) and salt (Lide and Kehiaian, 2020) phases are computed as
functions of temperature. Several options for composite thermal conductivity can be used; the default thermal conductivity
function is a linear scaling function of phase saturations:

$$\kappa = \kappa_{dry} + \phi \sum_{\alpha=l,g,h,i,s} s_\alpha \kappa_\alpha \tag{9}$$


where $\kappa_{dry}$ is the dry rock thermal conductivity and $\kappa_\alpha$ is the thermal conductivity of phase $\alpha$.

The presence of salt has several impacts on system behaviour. If present, salt precipitation reduces permeability, which affects
$CO_2$ injectivity, gas flow, and liquid imbition during injection and very far into the future. This occurs either at the injection





site if enough $CO_2$ is injected to dry out the water or far into the future when free phase $CO_2$ has undergone conversion to very high hydrate saturations (though at this point, permeability reduction due to salt precipitation is dwarfed by the presence of solid hydrate at high saturations). Aqueous dissolved salt concentration affects the density of the aqueous phase; the presence of gradients in salt concentration drives convective mixing. Salt exclusion during hydrate formation locally increases salt concentrations, which can produce this phenomenon (Fukuyama et al., 2023). Dissolved salt also affects gas solubility and shifts the three-phase equilibrium pressure of gas hydrate. Carbon dioxide equilibrium phase partitioning is computed using the method of Spycher and Pruess (2010); the $CO_2$ hydrate – free phase $CO_2$ – water three-phase equilibrium curve is determined from a polynomial fit of data from Men et al. (2022) up to the upper quadruple point (283K), beyond which point a steep line was used to preserve differentiability of the phase boundary.

**2.3 Phase States and Primary Variables**

PFLOTRAN's HYDRATE mode solves mass conservation, energy conservation, and well flux conservation equations for a set of three components ($CH_4$/$CO_2$/air, $H_2O$, NaCl) over five phases (aqueous, gas component-rich/gas, gas hydrate, ice, salt precipitate). This results in solving a set of four partial differential equations for all cells in the domain plus one coupled well equation per cell containing the bottom segment of a well. Therefore, PFLOTRAN's fully implicit solution solves for four primary variables everywhere plus one extra primary variable per well in the domain.

The reservoir (non-well) equations use primary variable switching depending on the thermodynamic state of a grid cell. HYDRATE mode contains 13 phase states with four primary variables per phase state (Table 1). For example, cells in the fully liquid (aqueous) saturated state solve for liquid pressure, dissolved gas mass fraction, temperature, and total salt mass per unit liquid mass as primary variables. Secondary variables like phase densities, viscosities, and enthalpies are computed at equilibrium from the primary variables through use of various equations of state. Precipitated salt saturation is computed by determining whether bulk salt concentration (total salt mass per mass of liquid phase) exceeds dissolved salt solubility and converting the excess salt mass into a solid phase (permeability updates according to Verma & Pruess [1988]). If dissolved gas mass fraction exceeds solubility and aqueous pressure, temperature, and dissolved salt mass fraction lie within the GHSZ (i.e., at pressures and temperatures above the 3-phase equilibrium boundary), the cell will transition into the hydrate-aqueous state and primary variables will update accordingly. Upon entering the hydrate-aqueous state, PFLOTRAN then switches primary variables and solves for gas pressure, hydrate saturation, temperature, and salt concentration.



**Table 1: Phase states and primary variable combinations in PFLOTRAN's HYDRATE mode**

| Phase State | Primary Variables | Phase State | Primary Variables |
|---|---|---|---|
| L (aqueous) | $P_l$, $x_l^g$, $T$, $m_l^s$ | AI (aqueous-ice) | $P_l$, $x_l^g$, $S_l$, $m_l^s$ |
| G (gas component-rich) | $P_g$, $P_a$, $T$, $m_b^s$ | HGA (hydrate-gas-aqueous) | $S_l$, $S_h$, $T$, $m_l^s$ |
| GA (two-phase gas-aqueous) | $P_g$, $S_g$, $T$, $m_l^s$ | HAI (hydrate-aqueous-ice) | $P_g$, $S_l$, $S_i$, $m_l^s$ |
| HG (hydrate-gas) | $P_g$, $P_a$, $T$, $m_b^s$ | HGI (hydrate-gas-ice) | $S_i$, $S_h$, $T$, $m_b^s$ |
| HA (hydrate-aqueous) | $P_g$, $S_h$, $T$, $m_l^s$ | GAI (gas-aqueous-ice) | $P_g$, $S_g$, $T$, $m_l^s$ |
| HI (hydrate-ice) | $P_g$, $S_h$, $T$, $m_b^s$ | HGAI (hydrate-gas-aqueous-ice) | $S_l$, $S_g$, $S_i$, $m_l^s$ |
| GI (gas-ice) | $P_g$, $S_i$, $T$, $m_b^s$ | | |

$P_l$ = liquid pressure, $P_g$ = gas pressure, $P_a$ = gas-rich gas (air) component partial pressure, $x_l^g$ = aqueous dissolved gas mass fraction, $T$ = temperature, $m_l^s$ = salt mass fraction per unit aqueous mass, $m_b^s$ = total salt mass per unit bulk volume, $S_g$ = gas saturation, $S_h$ = hydrate saturation, $S_l$ = liquid saturation, $S_i$ = ice saturation


For the well equation, the bottom hole pressure (BHP) of the well is solved as a primary variable. Given a user-defined well flow rate, each well's BHP is solved fully implicitly as part of the full reservoir flow solution. At a given BHP, the well model solves for all other pressures in the well by working upward and iteratively solving hydrostatic pressure of the injection fluid where variable fluid density in the well is updated as a function of pressure and temperature. Once the hydrostatic pressures

are determined, mass and energy fluxes between well and reservoir at the well segment centroids are computed by adding a hydrostatic adjustment to reservoir pressures to align reservoir pressures with well segment centroids and then employing Equation 7. Fluxes between coupled wells and reservoir cells are therefore functions of the well primary variable (BHP) and reservoir cell primary variables (e.g., gas/liquid pressure).

Using fully implicit coupling and a Newton-Raphson solution search method, insertion of a coupled well into the domain therefore adds additional fill to the Jacobian matrix used to compute solution updates. Critically, the well model adds extra connectivity beyond the typical stencil for two-point flux calculations. Thus, the structure of the Jacobian matrix is altered by introducing wells. This alteration is typically minimal but would likely become more severe as the number of wells in the domain is increased or as the number of uncased (i.e., screened) segments per well is increased. The number of wells, the

extent to which each well increases the fill of the Jacobian, and the strength of the coupling between well and reservoir are all likely to affect the overall performance of simulations using the coupled well model. PFLOTRAN uses neighbour cell ghosting to parallelize computations; the fully coupled well model updates the ghosting stencil to include all off process reservoir cells connected by a given well, allowing for consistent incorporation of well terms in the Jacobian both in parallel and in serial. When a well is turned off (or well flow rates are set to 0), that well equation is not solved.





## 3 Results

We demonstrate our developments by applying the software to a set of hypothetical $CO_2$ injection scenarios in marine environments within the GHSZ. In the first example, liquid $CO_2$ is injected slowly into a simple 1D homogeneous sediment column through a partially screened well beneath the GHSZ. The second example simulates commercial-scale injection of supercritical $CO_2$ into a 2D radial domain with heterogeneous layering, where a well is screened within and beneath the GHSZ. The $CO_2$ plume in this model is tracked for 10,000 years as it transitions from a supercritical phase to a dense liquid phase and then into the gas hydrate phase.

### 3.1 1D Liquid $CO_2$ Injection into a Homogeneous Reservoir

In this scenario, a relatively slow trickle injection is designed to illustrate the multiphase and thermodynamic processes associated with injecting $CO_2$ into the GHSZ. A 1D, 500-m homogeneous domain is initialized to hydrostatic conditions where the top of the domain is held at seafloor pressure of 10 MPa, seafloor temperature of 3º C, and geothermal gradient of 30º C/km. The top Dirichlet boundary condition is set to the initial seafloor temperature, pressure, zero gas mass fraction, and salinity (0.035 kg/kg). The bottom boundary at 500 mbsf is a Neumann zero flux boundary. The domain is discretized into 100 grid cells in the vertical dimension; individual grid cells measure 25 m in the horizontal x-dimension by 1 m in the horizontal y-dimension by 5 m in the vertical z-dimension. A well penetrates the entire domain and is uncased (i.e., screened) for 20 m from 475 meters below seafloor (mbsf) to 495 mbsf. Dense liquid phase $CO_2$ is injected at 20º C at an injection rate of 5,000 kg/yr for 150 years. Pressure of the injection varies along the well depending on the BHP, but $CO_2$ remains in the liquid phase for the entirety of the injection. Use of the well model will lead to variable $CO_2$ injection rates in each of the uncased well segments and correspondingly variable injection enthalpy as a function of well segment pressure; this effect is less noticeable in this homogeneous case than in the heterogeneous case. This simulation is run for 200 years. A constant reservoir porosity is set to 0.35, and constant isotropic permeability is set to $1\times10^{-13}$ m$^2$. A Van Genuchten capillary pressure function is used, where Van Genuchten $n$ is set to 1.84162, $\alpha = 0.5$ m$^{-1}$, and $S_{lr} = 0$. Corey relative permeability functions are used, where $m = 0.457$, $S_{lr} = 0.3$ and $S_{gr} = 0.05$.



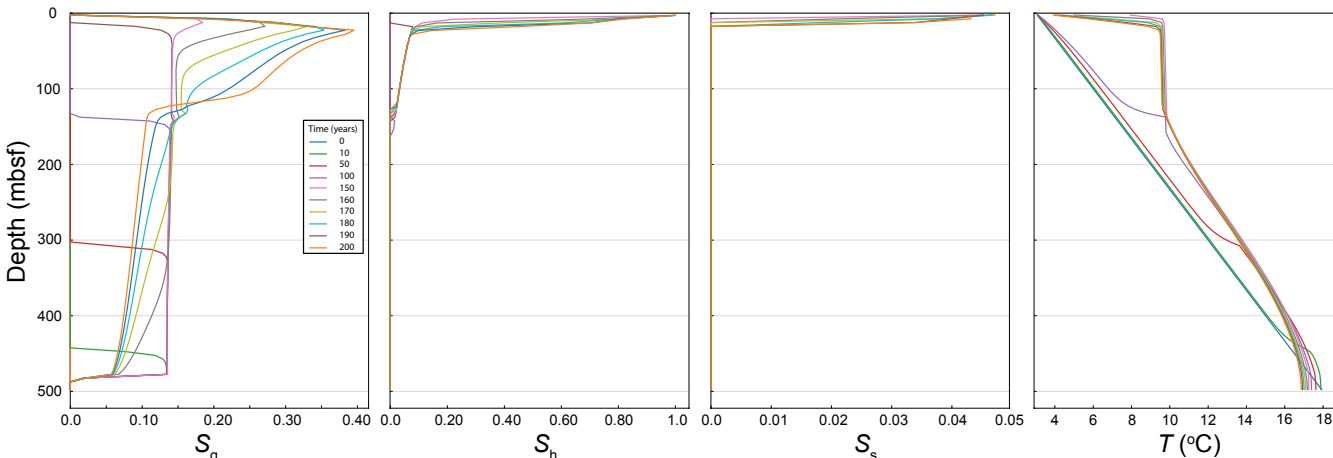

**Figure 1: Gas saturation, hydrate saturation, salt precipitate saturation, and temperature during and after a $CO_2$ injection beneath the GHSZ**

During the first 50 years of injection, the injected $CO_2$ remains beneath the base of the $CO_2$ hydrate stability zone (Figure 1). Therefore, it can only exist as free-phase $CO_2$. Thus, it migrates upward in characteristic fashion: buoyancy and pressure forces drive gas (free-phase $CO_2$) saturations exceeding the residual gas saturation to migrate upward in the sediment column. By 100 years, the free-phase $CO_2$ front has reached the base of the GHSZ. At this point, some free-phase $CO_2$ converts to $CO_2$ hydrate. Exothermic hydrate formation keeps the reservoir temperature at the three-phase equilibrium temperature while free-

phase $CO_2$ and hydrate coexist. As the gas plume migrates upward over time, more gas converts into gas hydrate. While the gas supply is strong and hydrate is still forming, the temperature of the reservoir is pushed well above the background (initial) geothermal temperature. $CO_2$ hydrate cannot form past approximately $10^{\circ}$ C, which is why the temperature throughout the three-phase zone is fixed to roughly $10^{\circ}$ C.

After the injection period ends, hydrate accumulates in high saturations near the top of the domain due to the seafloor temperature and pressure being fixed; the resulting permeability reduction causes gas to pool and salt to concentrate, leading to salt precipitation. This kind of scenario is unlikely to occur in a more realistic reservoir in 2D and 3D where permeability reduction would cause gas to migrate laterally and therefore not cause such significant pooling effects. But this model illustrates how merely relying on conversion of $CO_2$ to gas hydrate alone as a $CO_2$ trapping mechanism is likely insufficient due to the

thermal buffering effect of exothermic hydrate formation. Permeability reduction associated with gas hydrate formation can act to slow free-phase $CO_2$ migration, but, at least at early time, a combination of thermodynamic and other structural trapping mechanisms is likely necessary to ensure the long-term sequestration of most of the injected $CO_2$ in the subsurface GHSZ.





## 3.2 Supercritical CO₂ Injection into a 2D Heterogeneous Reservoir

In this scenario, a commercial-scale $CO_2$ injection is modelled under more realistic reservoir and injection conditions (Figure
2). A 2D, heterogeneous cylindrical domain extends from the seafloor down to 600 mbsf with a radius of 3.765 km. The model
domain consists of 40 grid cells in the horizontal dimension increasing in thickness from 7.38 m at the model centre to 364.36
m at the far edge. The model contains 55 cells in the vertical dimension with varying thickness, each corresponding to a
different layer in the model. The model consists of interbedded sand and mud units as might be found within the marine GHSZ.
High and low permeability layers alternate with synthetic heterogeneity; similarly, the model contains heterogeneous porosity
and capillary entry pressure. All other physical properties are kept constant between layers. A Brooks-Corey capillary pressure
function along with Burdine relative permeability functions for liquid and gas phases are used for all layers. For all layers,
Brooks-Corey $\lambda$ =0.8311 and $S_{rl} = S_{rg} = 0.0597$. The capillary entry pressure (the inverse of which is expressed by the Brooks-
Corey $\alpha$ parameter) varies between reservoir layers (Figure 3). Rock density is set to 2,650 kg/m³, dry rock thermal
conductivity is set at 2.0 W/m-C, and soil compressibility is modelled with a linear compressibility function using a soil
compressibility of 1.0E-8 Pa⁻¹ and a reference pressure of 10 MPa. Seafloor pressure is set to 10 MPa, seafloor temperature is
5° C, and seafloor salinity is 0.035 kg/kg. The model is initialized at hydrostatic pressure, constant salinity, and a geothermal
gradient of 20° C/km. The top and outer edges of the domain are kept at the initial conditions. The bottom boundary condition
is set to no liquid or gas flux, constant salinity, and a constant heat flux to preserve the geothermal gradient.

The well used in this scenario extends from the seafloor to 300 mbsf and is cased for the first 100 m. The rest of the well is
uncased; flow from well to reservoir is possible only in the uncased interval. Given the conditions outlined above, the bulk
BHSZ for $CO_2$ hydrate is at approximately 250 mbsf. Therefore, the well in this scenario extends through the GHSZ and 50 m
below the bulk BHSZ. Care was taken to ensure that the required well pressures to achieve the specified $CO_2$ injection rate
were realistic; the 100 m depth of well casing was chosen so that the well pressures resulting from our prescribed injection
rate did not exceed the lithostatic pressure. This calculation is approximate and does not consider the fracture gradient; the fact
that well pressures can easily approach the lithostatic gradient in these settings means that reservoir integrity should be
evaluated when performing site-specific evaluations of $CO_2$ injectivity in the shallow subsurface.



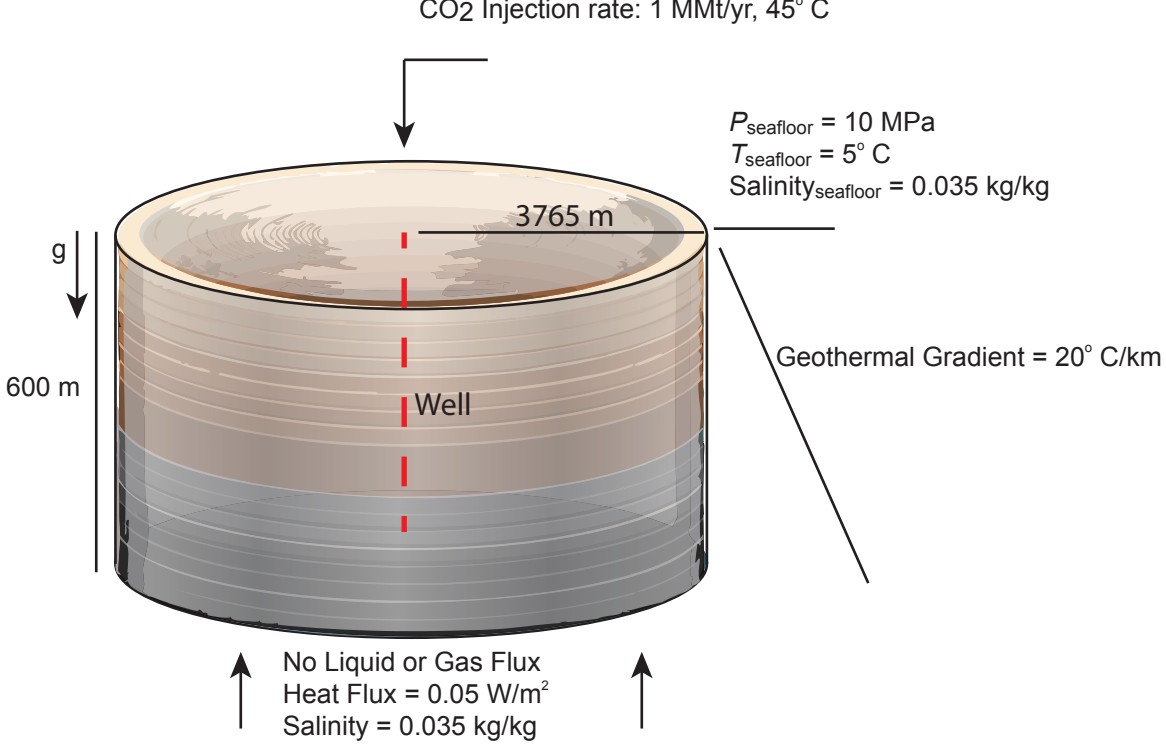

**Figure 2: Schematic of the heterogeneous 2D cylindrical model injection scenario**


The model is run for 10,000 years. The injection is designed as if the reservoir were a prospective DOE CarbonSAFE storage complex. Therefore, $CO_2$ is injected continuously at a rate of 1 million metric tons (MMT) per year for 50 years to meet a storage target of 50 MMT of $CO_2$. The $CO_2$ is injected at a constant temperature of 45° C; injection pressure will vary along the wellbore depending on the hydrostatic pressure of the well, and this in turn will affect the enthalpy of the injected gas.

After 50 years, the well is shut off; over time, warm supercritical $CO_2$ will cool into a dense liquid $CO_2$ phase and then eventually a gas hydrate phase.

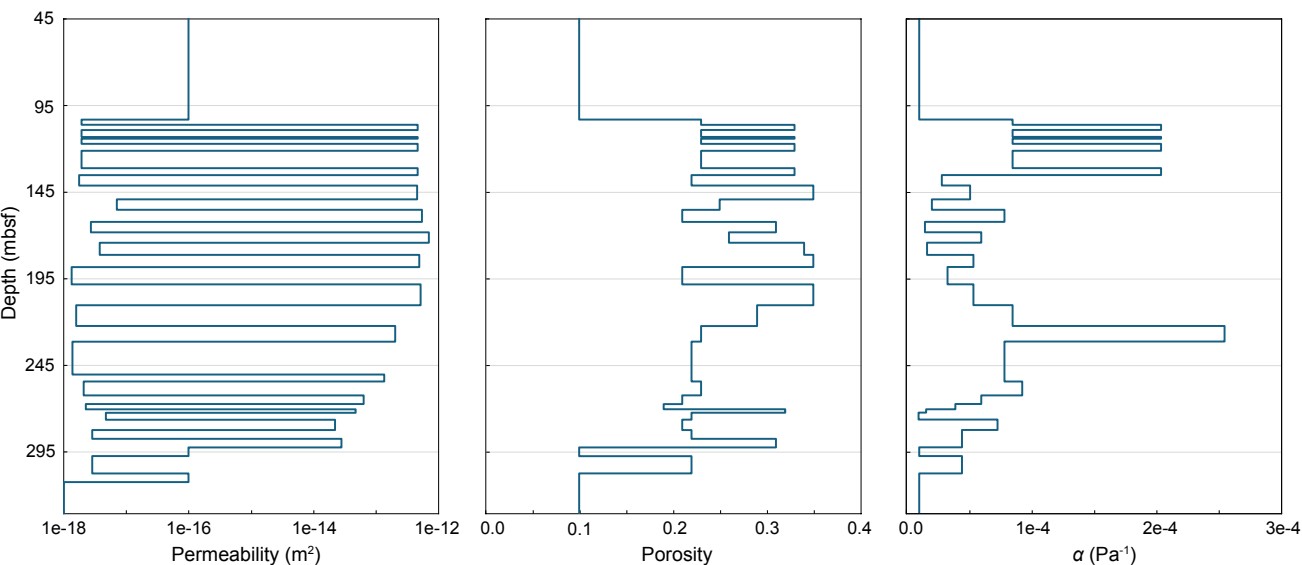

**Figure 3: Depth-varying physical properties of the heterogeneous 2D injection scenario, zoomed in near the injection interval.**
**Physical properties held constant above and below these snapshots.**

During the 50-year injection interval, gas flows predominantly into the high-permeability reservoir intervals (Figure 4, Figure 5). This is because the well model adjusts the gas flow rate (Figure 6) into individual intervals as a function of hydrostatic well pressure, reservoir pressure, and well index, where well index is a function of reservoir permeability. Since early hydrate
formation in the reservoir units elicits a permeability and pressure response, the well flow rate into individual units evolves over time during the injection. In some units, well flow rate drops, and these drops are then compensated by increases in flow rates in other units. Likewise, the pressure in the well evolves over time in response to hydrate formation and relative permeability of the mobile fluids.

By the end of the injection period, gas has flowed preferentially in the radial direction along high permeability flow paths. On the outer edges of the gas plume, free phase $CO_2$ combines with water to form a gas hydrate phase where pressures and temperatures are within the gas hydrate stability zone. Since pure $CO_2$ is being injected through the well and since water is miscible in the $CO_2$ phase, high gas saturations in the near-wellbore cells cause salt concentrations in those cells to increase above salt solubility. This salting out effect results in small amounts of salt precipitate saturation in the pore space at the end
of the injection.

**Figure 4: Snapshots of saturations over time in the vicinity of the wellbore. Depth is represented by the vertical axis, and radial distance from the well is represented by the horizontal axis. Gas (free-phase $CO_2$) saturation ($S_g$), hydrate saturation ($S_h$), and salt precipitate saturation ($S_s$) distribution at 50 years, 1,000 years, and 10,000 years of simulation time. A zoomed-in cut-out shows near-wellbore salt precipitate saturations at 50 years.**

As water imbibes back into the near-wellbore cells between 50 and 1,000 years, gas saturations in those cells drop and salt re-dissolves (salt precipitate saturations near the wellbore drop toward 0). During this time, the temperature of the injected fluid is dropping toward the background temperature field. As this happens, free phase $CO_2$ combines with available water to form gas hydrate. Exothermic hydrate formation props up temperatures during hydrate formation and slows the process of $CO_2$ conversion into gas hydrate. In some areas at the upper edges of the $CO_2$ plume, where the system is furthest into the GHSZ, very high conversion of $CO_2$ to gas hydrate is achieved in a relatively short amount of time. Since hydrate formation only involves water and $CO_2$ components, salt exclusion during rapid hydrate formation results in local buildup of salt





concentrations. Some cells in the model associated with rapid hydrate formation therefore exhibit some solid salt precipitation

by 1,000 years.

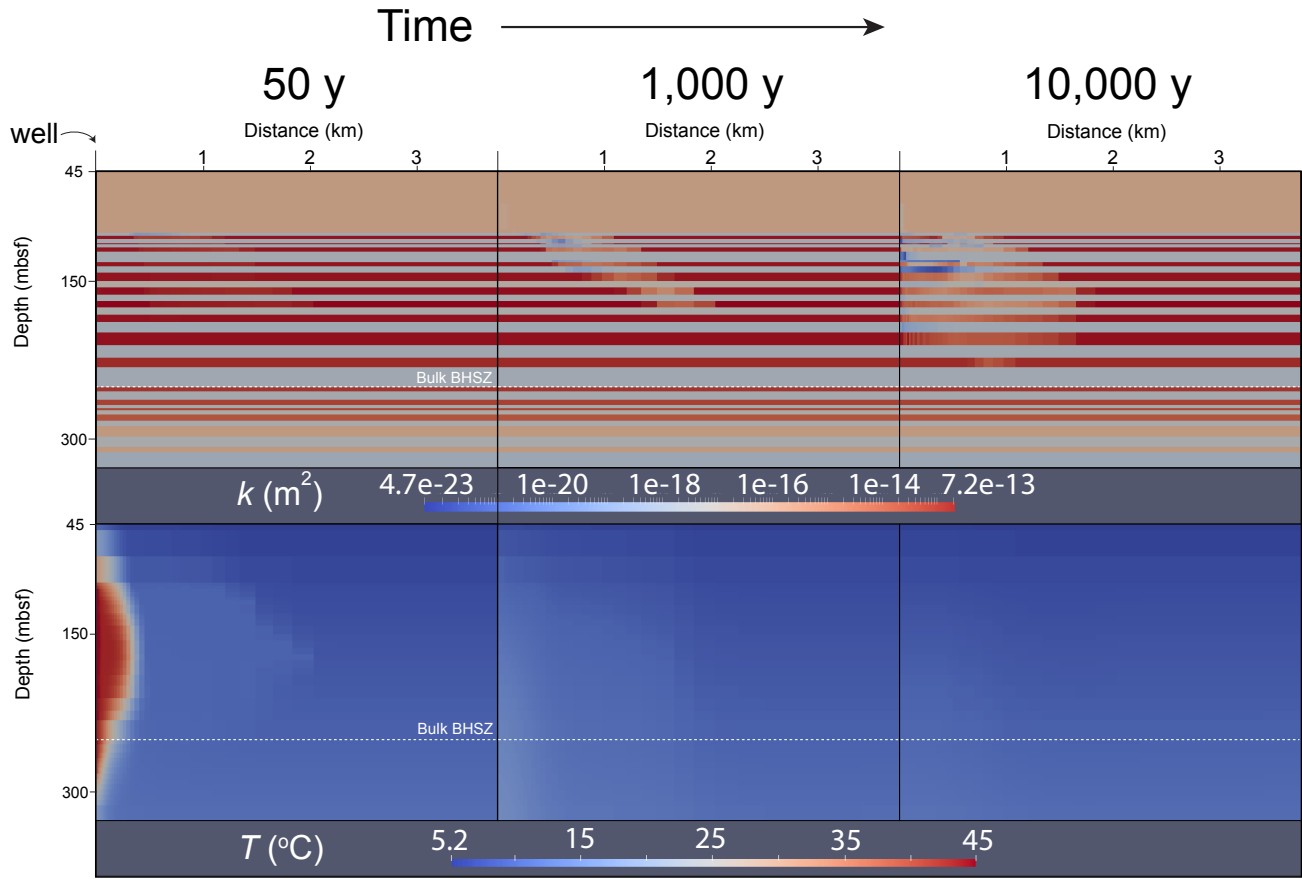

**Figure 5: Permeability (*k*) and temperature (*T*) distribution at 50 years, 1,000 years, and 10,000 years of simulation time. Depth is represented by the vertical axis, and radial distance from the well is represented by the horizontal axis.**

After 10,000 years, most of the injected $CO_2$ has converted into gas hydrate. High gas hydrate saturations have built up in the

near-wellbore area since the initial temperature of the injection has decayed away toward the steady-state geothermal

temperature profile. Hydrate formation has significantly decreased the permeability of the host reservoir, and gas has migrated

into the other layers to form hydrate. A region of three-phase coexistence (liquid water, free phase $CO_2$, and gas hydrate) is

still present after 10,000 years because of a combination of exothermic hydrate formation and slow imbibition rates of cool

liquid water due to significant permeability reduction.


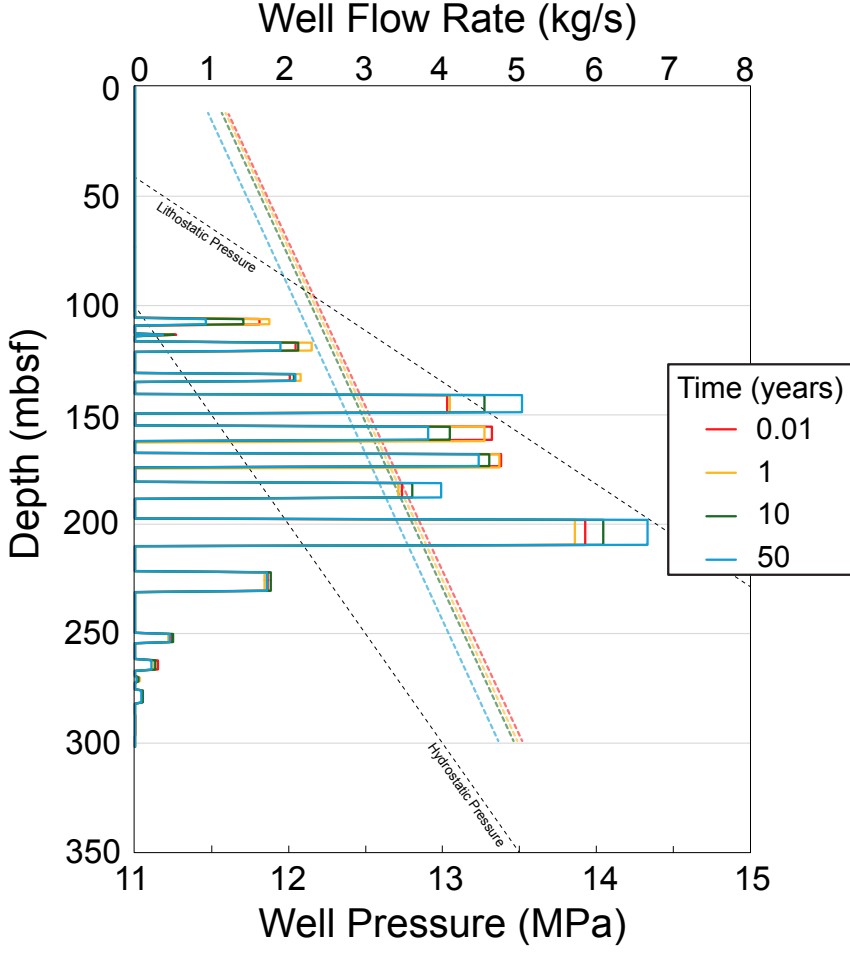

**Figure 6: CO₂ mass flow rate in each well segment (solid lines) and supercritical CO₂ phase pressure in the well (dashed lines) during the 50-year injection period.**

**Discussion**

The two models presented here were selected to demonstrate some of the key dynamic coupled processes associated with CO₂ injection into the gas hydrate stability zone. In the 1D homogeneous model, CO₂ is injected beneath the GHSZ and forms a free phase which migrates upward due to buoyancy and pressure forces. Once it enters the GHSZ, conversion of CO₂ into an immobile hydrate phase is limited by the rate at which heat can diffuse away and availability of water. When thermal conduction and liquid water flow are limited, the system can maintain 3-phase equilibrium temperature for decades (or

thousands of years as is shown in the 2D model). This thermal buffering phenomenon has been observed in models of natural methane hydrate formation and dissociation in marine sediments and can occur on geologic timescales depending on free gas





phase flow rate or rate of environmental change (You and Flemings, 2018; Oluwunmi et al., 2022). Clearly, such a scenario would not be ideal for permanent $CO_2$ sequestration as much of the $CO_2$ remains in a free phase and accumulates very close to the seafloor. Permeability reduction due to hydrate formation adds a physical trapping mechanism analogous to a low

permeability sealing facies. The fact that this permeability reduction is the primary mechanism for preventing $CO_2$ flow to the surface in the 1D model suggests that physical/structural trapping should be considered just as important or more important than thermodynamic trapping when evaluating a reservoir within the GHSZ for long-term $CO_2$ storage.

The 2D cylindrical model was designed to incorporate more realistic reservoir physical properties and include an injection rate

more viable for commercial-scale $CO_2$ storage in the GHSZ. In this scenario, $CO_2$ was injected into a layered reservoir that is bounded by low permeability facies that inhibit direct flow of $CO_2$ to the seafloor. Instead of injecting beneath the GHSZ, a high-temperature supercritical $CO_2$ phase is injected directly into and directly beneath the GHSZ. Near-wellbore gas hydrate formation is prevented by the high temperature of the injection during the injection period. Hydrate formation does occur during the injection period at the edges of the $CO_2$ plume; the associated changes in fluid mobility and permeability alter the

pressure in the well and cause well flow rates to fluctuate layer-by-layer. Therefore, even if the $CO_2$ injection temperature is designed to prevent near-wellbore hydrate formation, hydrate formation in the far-field should be considered when designing a $CO_2$ injection insofar as it could appreciably affect wellbore pressure. Salt precipitation can occur near the wellbore during injection due to "salting out" effects of dry $CO_2$ injection. Salt can also precipitate later in time as $CO_2$ converts to hydrate faster than the pore water can freshen through either aqueous imbibition or salt diffusion. In either case, salt precipitate

saturations appear to be minimal for the scenario modelled here, but salt precipitation could appreciably decrease permeability under a configuration with either more rapid $CO_2$ injection or more rapid conversion of $CO_2$ to hydrate. In some regions of this model, hydrate saturations become very high at late times and lower the permeability of host reservoir units by several orders of magnitude. This makes for effective sealing of $CO_2$ by conversion to an immobile phase and by impeding flow of the free $CO_2$ phase. This phenomenon also has the effect of pushing gas into less intrinsically permeable layers and ultimately

smoothing the distribution of gas hydrate throughout the model domain.

**Conclusions**

We present several new developments in PFLOTRAN's HYDRATE mode including a new option to model $CO_2$ as the working gas, a new salt mass balance for considering effects of salinity gradients and salt precipitation, and a new fully coupled hydrostatic well model. We demonstrate these new capabilities on a series of test problems designed to explore the coupled

processes relevant to $CO_2$ injection into the marine gas hydrate stability zone for the purpose of permanently sequestering $CO_2$. $CO_2$ sequestration in the gas hydrate stability zone is a potentially promising technique for secure storage of $CO_2$ because of the associated favourable conditions for converting injected $CO_2$ into solid gas hydrate form, which is immobile in the pore space. However, no reservoir modelling studies to date have demonstrated what commercial-scale $CO_2$ injection into the gas

hydrate stability zone might look like. We show through a 1D homogeneous model that it is critical to consider multiple
trapping mechanisms in addition to the thermodynamic trapping accompanied by conversion of $CO_2$ into hydrate form. We
then expand to a 2D heterogeneous cylindrical model with a commercial-scale 1 MMT/yr $CO_2$ injection rate to underscore the
interplay between structural trapping, thermodynamics, and permeability alteration on the migration and conversion of $CO_2$.
We demonstrate how our fully implicit well model adapts to changes in flow properties during $CO_2$ injection, and how injection
of a warm supercritical $CO_2$ phase can facilitate near-wellbore injectivity but can lead to pressure change in the well. In the
future, this capability could be used to more rigorously evaluate the potential for secure $CO_2$ storage in greater volumes, at
larger (3D) scales, with more site-specific inputs, and with more exotic well designs including multiple wells or horizontal
wells.

## Code and Data Availability

The software developments described here were released on August 23, 2024 with PFLOTRAN version 6.0
(www.pflotran.org). PFLOTRAN is open source and freely available under a GNU LGPL Version 3 license at
https://bitbucket.org/pflotran/pflotran. Software inputs and a snapshot of the PFLOTRAN v6.0 Bitbucket repository are
available on Zenodo at https://zenodo.org/records/13619874. The files on Zenodo include PFLOTRAN input decks for both
model scenarios and associated Span-Wagner EOS database files referenced by those input decks.

## Acknowledgements

This research was supported by the U.S. Department of Energy (DOE) Office of Fossil Energy and Carbon Management and
the National Energy Technology Laboratory (NETL), Award No. FWP 72688. This work was also supported by Pacific
Northwest National Laboratory's (PNNL) Laboratory-Directed Research and Development (LDRD) program, Award No.
211622. PNNL is operated for the DOE by Battelle Memorial Institute under contract DE-AC05-76RL01830. This paper
describes objective technical results and analysis. Any subjective views or opinions that might be expressed in the paper do
not necessarily represent the views of the U.S. Department of Energy or the United States Government. Generative AI was
used to develop part of the schematic illustration in Figure 2.

## Author Contribution:

MN: software development, model conceptualization, formal analysis, methodology, writing; JB: model conceptualization,
writing; FN: model conceptualization, writing; GH: software development, review & editing.

**Competing Interests:** The authors declare that they have no conflict of interest.



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
