# Peer review of "Modeling Commercial-Scale CO2 Storage in the Gas Hydrate Stability Zone with PFLOTRAN v6.0"

_Geoscientific Model Development, 2024_

## Author Response (AR1)

RC1:

This paper presents the development of the HYDRATE flow model within the PFLOTRAN framework, designed to simulate $CO_2$ storage in the form of $CO_2$-hydrate. The model is applied to assess two distinct scenarios of $CO_2$ injection into the hydrate stability zone. This modeling approach provides valuable insights into the potential challenges and opportunities associated with commercial-scale $CO_2$ storage as hydrate. The manuscript is well-constructed and includes high-quality figures. I recommend the following minor revisions to enhance its clarity:

1. The description about well model is not quite clear. How is the well pressure calculated? I suggest a separate section describing the well model.
   Agreed. We have moved the description of the well model into a separate section, added more detail on the calculations that go into the well fluxes, and added more description of the well pressure calculation (Section 2.2).

2. Please briefly explain "Neighbor cell ghosting" in the text.
   Thank you, this was not clear. We have added a more thorough description of cell ghosting and how this is affected by introduction of a well (Lines 305-320).

3. Please add the depth of BHSZ in section 3.1
   The initial depth of the BHSZ has been added to Figure 1, along with some additional description in the figure caption. The BHSZ is altered over time by upward migration of warm $CO_2$.

4. I suggest adding a phase diagram used in this study.
   Good suggestion, thank you. We have added the phase diagram (Figure 1) and expanded some associated referencing.

Thank you for your review and for the suggestions. They have significantly improved the manuscript.

RC2:

The authors provided well-written and concise modeling study on two CO2 injection scenarios into marine sediments within the gas hydrate stability zone. Please find below my suggestions on how to further improve the content of this paper related mostly to the modeling scenarios description and presentation of the results.
Detailed comments:
- line 10: 'prospective storage sites tend to be terrestrial' - this is probably true from the US perspective, however, the entire sentence on structural, petrophysical etc. trapping is absolutely right also for marine settings. I would therefore recommend to remove the terrestrial part.
Thank you, great point. This language has been removed.
- line 14: I would say upfront whether you refer to the CO2 GHSZ or CH4 GHSZ which might be much more common to the readers
Good point. Where we introduce the acronym GHSZ, we have clarified that this refers to the $CO_2$ hydrate stability zone.

- line 23: 'terrestrial' CO2 storage - again, perhaps the 'terrestrial' part could be removed as it applied to all types of sites

This has been fixed. Thank you.

- line 39: if you use CO2 throughout the text, then also use CH4 instead of methane

Agreed. This has been updated.

- line 41: no need to have both terms- carbon dioxide and CO2 here, it was stated already in the abstract

This has been removed.

- line 42: 'CO2 hydrate that forms in similar pressures and temperatures...' - I would say that these stability differences between CH4 and CO2 are rather crucial? To help the readers, you could give some examples on GHSZ thickness for both components at identical conditions.

Yes, the difference between the CH4 hydrate stability zone and the CO2 hydrate stability zone is important. While we consider CH4-free sediments in this study, the simulator can simulate CH4 hydrate formation. The CH4 hydrate phase boundary has been included in the newly added Figure 1 for readers to discern where both hydrates are stable.

- line 46: please stick to CH4 instead of methane for consistency

Done. Thank you.

- line 48: 'to kick out methane' is a bit of a jargon

Thanks. This has been reworded.

- line 49: what about released CH4 during this substitution process? Could you model this process as well?

At the moment, we can model either CH4 or CO2 hydrate, not substitution.

- line 56: 'Relative to deeper sediment, this point usually represents a minimum temperature and pressure' - please rephrase or remove this sentence

Removed.

- lines 66-67: repetition of 'demonstrated'

Done.

- line 134-135: 'This means that for one well, only one extra row and one extra column are used in the fully implicit flow Jacobian, not an extra row and extra column for each reservoir cell associated with a well' - please rephrase the sentence for clarity or add some additional description

This sentence has been cleaned up, and additional description has been added.

- line 156: 'salt is tracked in the aqueous phase' - since there is salt precipitation in the system, you could explain the salt mass balance in a bit more detail here

Thank you, this was unclear. Salt can be both in the aqueous and salt precipitate phases.

- line 218 and 220: '...and very far into the future.' - please explain a bit more precisely the time frame

This statement has been modified to clarify the two scenarios that can lead to salt precipitation: dry-out during CO2 injection and rapid hydrate formation excluding salt.

- line 286: 'seafloor temperature' - do you mean bottom water temperature? is it realistic to assume a constant bottom water temperature for 150 years?

The text has been altered to refer to bottom water temperature instead of seafloor temperature. And yes, bottom water temperature would likely vary over this time period. It is possible to apply a time-dependent temperature boundary condition in PFLOTRAN, but for this study we chose to highlight the thermal effects of a $CO_2$ injection and refrain from generating a synthetic time-dependent bottom water temperature. But a site-specific investigation could apply a bottom water temperature projection.

- line 291: how realistic is the injection rate of 5,000 kg/year for 150 years? Why did you use these exact numbers?

Thank you for pointing this out. This has been revised to a 50 year injection of 15,000 kg/yr to be more consistent with Example 2.

- line 292: what is 'BHP'?

Bottomhole pressure of the wellbore. Since this isn't a common acronym, we've removed the acronym.

- line 305: please indicate the base of GHSZ in fig. 1

Done.

- line 320: 'CO2 trapping mechanisms insufficient due to thermal buffering effect' - when would the buffering effect work? what would be the critical conditions e.g. injection rates or temperature of CO2? How long would it take to equilibrate the system? Some discussion needed

This point was unclear, so we have added some clarification. The point was to show how forming hydrate releases heat and therefore can preserve free-phase CO2 mobility through a zone of three-phase equilibrium. Example 2 was designed to show the timescales of thermal dissipation through a more realistic scenario.

- lines 336-337: why in scenario 2 dTdz and Ttop are different than in scenario 1 (20/30 deg.C/km, 3/5 deg.C)?

The idea here was to present a 1D simulation showing buoyant CO2 migration and exothermic hydrate formation in a homogeneous medium. The intent behind using different model parameters was to show some diversity in model inputs, but we agree that making the models too different can be confusing. We have simplified some of the parameters, made the thermal gradient the same between the 1D and 2D simulations, and extended the bottom of the model domain. Hopefully this reduces some of the confusion.

- line 351: what is 'DOE CarbonSAFE storage complex'?

This was the motivation for choosing a 1 MMT/yr injection rate for 50 years and was mentioned in the introduction. Referencing this here likely adds confusion, so we've removed the reference.

- line 362: 'well model adjusts the gas flow rate' - what do you mean? is your simulations accounting for that or is it physically happening during injection experiments? please explain.

This is referring to how injected CO2 is distributed at the injection point as a function of reservoir permeability and well pressure, as shown in Figure 7. This statement has been cleaned up to make this clearer.

- lines 374-375: how does salt precipitation affect your modeled migration fields for both gas and pore fluids? I would suggest to include the velocity fields either in one of the existing figures or as a separate material. Your discussion also refers a lot to the

permeability reduction effect (which is nice) which could be more highlighted in the paper as one of your key results.

Thank you for the suggestion. We have added supplementary videos showing permeability evolution and velocity field evolution.

- line 442: ' we demonstrate these new (model) capabilities on a series of test problems' - in the current paper, there are two scenarios which might seem not sufficient for this term. I would rather explicitly say '(...) on two test problems', just for clarity.

Agreed. This has been changed.

Figures:
- in general, GHSZ should be indicated on all figures. Fig. 1 Sh and Ss could be presented in a zoom since non-zero concentrations are only in the uppermost part.

Thanks for the suggestion. The initial BHSZ has been added to all figures. We decided to keep all plots on the same depth range for ease of viewing, but in light of the questions about the salt mass we decided a plot of dissolved salt was more instructive of how salt mass was being updated. Though salinity variations are fairly minimal in this 1D model, salt mass fractions spike where hydrate formation has consumed significant amounts of water.

- perhaps it would be a good idea to compile a movie from each injection scenario as a supplementary material

We have added videos to the supplement. Thanks for the suggestion.

- Fig.2 - please mark intervals from fig. 3

The layering in the figure was meant to illustrate heterogeneous layering in a generic sense, with labels and the subsequent figure indicating specific model parameters.

- Fig. 3: what is alpha (Pa-1) in the last panel?

This is the inverse of the capillary entry pressure, which parameterizes the capillary pressure model. A description of variables has been added to the figure caption

A suggestion for enhanced content: I would be glad to see a discussion on the implications of multi-phase migration within sediment overburden above the well/injection points which might lead to a potential leakage, and your predictions on challenges related to the injection inside a mixed CO2-CH4 GHSZ with pre-existing CH4-GH deposits.

We have added some discussion of $CO_2$ leakage and the potential implications of CO2 injection into the GHSZ with existing hydrates.

Thank you for your detailed review; we greatly appreciate the feedback and believe it has improved the manuscript significantly.

---

## Author Response (AR2)

Dear Editor,

Thank you for reviewing our revised manuscript. We appreciate your comment about the legibility of some figures. The following comments have been addressed (responses in red):

1. Thank you for addressing the comments and suggestions from the reviewers. Before accepting your manuscript for publication, I would however like you to improve slightly the figures. Namely, it would be ideal if you could add letters to sub-panels such that it would be clearer and simpler to refer to those from the text and from within the figure caption. You may further want to slightly increase the axis label font size. Lastly, it would be ideal if you could enhance the captions such that it would refer to specific sub-panels and also make sure to describe the variables used in the plots.

   We have updated Figures 5 and 6 with sub-panel lettering and updated the figure captions accordingly. We have also updated the caption in Figure 1 to be more descriptive and the caption in Figure 2 to reference the variables in the figure. We have increased label font size in Figures 1, 2, 4, 5, and 6 to be more legible.

Please let me know if you need anything else.

Michael Nole